# Numerical Study on a Liquid Cooling Plate with a Double-Layer Minichannel for a Lithium Battery Module

**DOI:** 10.3390/mi14112128

**Published:** 2023-11-20

**Authors:** Yu Xu, Ruijin Wang

**Affiliations:** School of Mechanical Engineering, Hangzhou Dianzi University, Hangzhou 310018, China; xuyude@hdu.edu.cn

**Keywords:** liquid cooling plate, lithium battery module, maximum temperature, temperature difference, minichannel heat sink

## Abstract

The liquid cooling system of lithium battery modules (LBM) directly affects the safety, efficiency, and operational cost of lithium-ion batteries. To meet the requirements raised by a factory for the lithium battery module (LBM), a liquid cooling plate with a two-layer minichannel heat sink has been proposed to maintain temperature uniformity in the module and ensure it stays within the temperature limit. This innovative design features a single inlet and a single outlet. To evaluate the performance of the liquid cooling system, we considered various discharge rates while taking into account the structure, flow rate, and temperature of the coolant. Our findings indicate that at a mass outflow rate of 20 g/s, a better cooling effect and lower power consumption can be achieved. An inlet temperature of 20 °C, close to the initial temperature of the battery string, may be the most appropriate because a higher temperature of the coolant will cause a higher temperature of LBM, so far as to exceed the safe threshold value. In the case of larger rate discharge, the design of a double-layer MCHS at the bottom and an auxiliary one at the side can effectively reduce the maximum temperature LBM (within 28 °C) and maintain the temperature difference in the single cell at approximately 4 °C. In the case of non-constant discharges, the temperature difference between cells increases with the maximum temperature. When the discharge rate is reduced, the large temperature difference helps the temperature to drop rapidly. This can provide guidance for the design of cooling systems for the LBM.

## 1. Introduction

With the rapid growth of vehicles, the pressure of energy shortages and environmental pollution has increased. From the perspective of sustainable development, the development of electric vehicles is an effective solution [1,2]. Lithium-ion batteries are particularly crucial as a source of energy for electric vehicles. The appropriate operating temperature range for lithium batteries should be controlled within the range of 20–40 °C [3,4], and the temperature difference between cells should be less than 5 °C. As the battery module is encased by an outer protective shell and has a compact internal structure, it is prone to heat accumulation inside the battery module. When designing a battery thermal management system (BTMS) for new energy vehicles, it is crucial to ensure both the effectiveness of the heat dissipation system and the uniformity of temperature among cells [5,6].

The cooling methods of BTMSs include air cooling, liquid cooling, and phase change material cooling (PCM), depending on the cooling medium [7,8,9]. Compared to air cooling, liquid cooling systems have a more complex structure, larger overall mass, and higher production costs. However, liquid cooling systems have a compact structure and provide a better cooling effect [10]. Wu et al. [11] developed and constructed a battery system utilizing direct liquid immersion cooling. The battery system was tested at charge rates of 0.5C and discharge rates of 1C, resulting in temperature differences of 4.9 °C and 8.8 °C, respectively. These temperature differences ensured that the maximum temperatures remained within the optimal working temperature range. Xin et al. [12] proposed a hybrid BTMS based on air and liquid cooling, where the heat generated by the battery is transferred to the coolant through uniformly distributed heat conductors along its axis to maintain the normal operation of the battery pack. Air cooling is then introduced to maintain battery temperature homogeneity at the edge of the battery pack. PCM cooling is a novel cooling method known for its compact structure and high cooling efficiency [13]. However, one drawback is that the heat absorbed by PCM cooling is not easily distributed to the outer parts of the battery pack. This can lead to thermal saturation and limit its application in production due to the high production cost [14]. To address this issue, Yang et al. [15] proposed a hybrid liquid cold plate that combines cooling channels and PCM for BTMSs. The hybrid cold plate has shown promising results, with more than 50% reduction in pumping power and an average temperature below 40 °C.

As the endurance capacity of electric vehicles continues to increase, there is a growing demand for high-power, fast-charging power battery systems. In order to meet the working requirements under high-rate charging and discharging conditions, the mainstream pure electric models on the market utilize liquid-cooled battery packs for efficient heat dissipation [16,17]. MCHSs have found extensive applications in microelectronics, aerospace, high-temperature superconductors, and various other fields. This is mainly attributed to their compact structure, high heat exchange efficiency, lightweight design, and reliable operation [18,19]. Xie et al. [20] focused on optimizing the channel structure of LCPs to enhance thermal performance. They presented the importance of designing the layout of LCPs and modifying the microchannel structure in order to improve heat dissipation capacity in [21,22]. 

The layout of the LCP depends on the geometry of the cell. There are three basic battery shapes on the market: cylindrical, square, and pouch. Lai et al. [23] designed a lightweight, space-efficient structure that can simultaneously contact and exchange heat with three cylindrical cells. Chen et al. [24] set up a proxy model aiming at thermal performance and energy consumption by placing an LCP with a parallel channel structure at the bottom of the square battery module and found that the height of the minichannel significantly influences the cooling effect, temperature uniformity, and energy consumption. Deng et al. [25] proposed a double-layer LCP with bifurcated flow channels consisting of a collecting layer channel and a dispersive layer channel. This optimal design aims to reduce the maximum temperature, standard deviation of the surface temperature, and pressure drop in the LCP simultaneously. It is important to note that the channel structure in liquid cooling systems also plays a significant role in influencing the cooling performance. Wang et al. [18] presented four approaches that can enhance heat transfer in the micro/mini-channel, destruction of the boundary layer, elevation of chaotic convection, enlargement of heat transfer area, and coolant with higher thermal conductivity. Wang et al. [26] designed an MCHS with periodically arranged V-ribs that can interrupt the thermal boundary, induce chaotic convection, increase the heat transfer area, and subsequently improve the heat transfer performance. Kalkan et al. [27] discovered that utilizing an LCP structure with minichannels, instead of the conventional serpentine channel liquid cooling plate, led to a significant decrease in the maximum temperature of the battery pack and a 40% improvement in temperature uniformity among the individual monomers, when subjected to varying discharge rates. Similarly, E. et al. [28] determined that cooling performance was primarily influenced by the number of channels and flow rate. They concluded that an appropriately sized structure can effectively enhance cooling capacity and minimize energy consumption. Wang et al. [29] arranged circular tubular channels in silicon plates for indirect liquid cooling of the battery and found that the number of channels determines the uniformity of battery temperature, and flow direction has little influence on cooling performance. Ding et al. [30] discovered that the rectangular channel exhibited superior heat dissipation and reduced the maximum temperature of the battery in comparison to the circular channel. Zhang et al. [31] investigated the impact of rectangular, trapezoidal, and circular flow channels on system cooling performance and determined that the trapezoidal channel possessed optimal cooling capacity. Huang et al. [32] found that a streamlined channel structure can effectively reduce the pressure drop in liquid cooling systems, ensuring uniform battery temperature and improving system cooling capacity. Additionally, external forces such as magnetic and electric fields can enhance heat transfer performance by inducing chaotic convection [33,34,35]. Du et al. [33] indicated that chaotic convection due to the Kelvin force, as well as the thermophoretic effect, can enhance heat transfer. Wang et al. [34] illustrated that the main mechanism for heat transfer in MCHS is chaotic convection induced by the motion of electrophoresis and thermophoresis. In the end, several publications [36,37,38] reviewed the thermal management of LBM and the design improvement and optimization of LCP for electric vehicles.

The performance of a liquid cooling system can be affected by multiple factors, including the temperature of the coolant, the flow rate, and the ambient temperature. In this study, LCP with a double-layer MCHS is designed to regulate the battery temperature in response to the heat generated during battery operation according to the specifications raised by a factory from Anhui Province, China. To meet its applicability, the simulation verifies that it can effectively ensure the uniformity of module temperature by controlling the flow rate and temperature at different discharge rates. When discharging at a high rate, the temperature of the upper part of the battery can be reduced by increasing the auxiliary LCPs to improve the heat dissipation performance. These results can provide valuable reference modules for future development.

## 2. Geometric Model

The main task of the present work is to design an LCP for lithium battery packs containing tens or hundreds of individual cells. Hence, the structure of the battery pack is simplified, and only the corresponding cells, thermal pads, and liquid cooling plates are retained. The employed battery is assumed to be a square ternary lithium battery. The thermal conductivity of a single cell is anisotropic, and thermal effects due to the positive and negative electrodes of the cell can be neglected. The cell specifications are listed in Table 1. The material for the LCP is aluminum alloy, and the coolant is water–glycol (50–50%). The thermal and physical parameters of the LCP, coolant, and thermal pads are provided in Table 2. 

The LBM comprises 15 cells that are connected to a liquid cooling plate through a thermal pad. These cells are assumed to have anisotropic thermal conductivity, with lower conductivity in the thickness direction. Under different discharge conditions, the heating power of the cell is different; thus, two different LCP structures are proposed. Figure 1 illustrates the layout of bottom cold plates with a double-layer (DLCP) and single-layer (SLCP), respectively. The minichannel structures in DLCP with one inlet and one outlet are shown in Figure 1a, and those of SLCP with two inlets and two outlets are shown in Figure 1b. The inlets and outlets for SLCP and DLCP are arranged at the center of the LBM. To enhance heat dissipation, fluid is introduced near the symmetry line. Since LCPs are not allowed at the top of the LBM due to the replacement operation, the top of the battery is prone to heat accumulation, and the heat at the top can be reduced by arranging side LCPs (see Figure 2a). The thickness of the aluminum alloy material is 1 mm, and the height of the minichannel is 3 mm and 1.5 mm for the bottom LCP and side ones. The length, width, and height of the bottom SLCP and DLCP are 420 mm × 148 mm × 5 mm and 420 mm × 148 mm × 9 mm, respectively. Additionally, the size of the side LCP is 420 mm × 46.5 mm × 3.5 mm. Moreover, the inlet channel widths are 30 mm, 15 mm, and 15 mm for the bottom SLCP, DLCP, and side LCP, respectively. 

To investigate the cooling performance of LCP in the LBM, various monitoring points are set up to observe temperature changes over time at different locations. These monitoring points are strategically positioned in a non-equidistant manner, considering the improved heat dissipation near the cold plate, to better explore the temperature variations across the cell. Figure 2b illustrates that T1, T4, T8, T11, and T15 are placed at the same position in the LBM for different cells to monitor the temperature between the two cells. Additionally, vertical monitoring points, Tv1–Tv5, are arranged at different locations on the middle cell’s side.

## 3. Numerical Analysis

### 3.1. Numerical Model

To simplify the simulation, the following assumptions can be made [40]: (1)The cold plate is homogeneous and isotropic.(2)With the exception of the upper surfaces, the other five surfaces of the cold plate are assumed to be adiabatic.(3)Single-phase, incompressible, and steady flow are assumed.(4)The thermophysical properties of fluid and solid are independent of temperature.

The governing equations for the coolant can then be written as follows. Conservation of mass (continuity) equation:(1)$$\frac{\partial \rho_{l}}{\partial t}+\frac{\partial (\rho_{l}u)}{\partial x}+\frac{\partial (\rho_{l}v)}{\partial y}+\frac{\partial (\rho_{l}w)}{\partial z}=0.$$

Momentum conservation equation:(2)$$\rho_{l}\left[\frac{\partial \vec{v}}{\partial t}+(\vec{v}\cdot \nabla )\vec{v}\right]=\rho_{l}f-\nabla P+\mu \nabla^{2}\vec{v}.$$

Energy conservation of fluids:(3)$$(\frac{\partial }{\partial t}+\nabla \vec{v})(\rho_{l}C_{p,l}T_{l})=\nabla (k_{l}\nabla T_{l}),$$
where $\rho_{l}$ is the density of the coolant, $\vec{v}$ is the flow velocity, P is the pressure of the coolant, μ is the dynamic viscosity, $k_{l}$ is the thermal conductivity of the coolant, $C_{p,l}$ is the specific heat capacity of the coolant, and $T_{l}$ is the temperature of the coolant. 

For the square lithium cell, the cell is assumed to be a homogeneous heat source, and the effects of internal convective heat transfer and radiation are neglected. Moreover, the density, specific heat capacity, and anisotropic thermal conductivity of the cell do not change with temperature, and the internal resistance of the cell does not change with temperature and discharge depth during discharge and remains at a constant value [41].
(4)$$\rho_{b}C_{p,b}\frac{\partial T_{b}}{\partial t}=k_{b,x}\frac{\partial^{2}T_{b}}{\partial x^{2}}+k_{b,y}\frac{\partial^{2}T_{b}}{\partial y^{2}}+k_{b,z}\frac{\partial^{2}T_{b}}{\partial z^{2}}+Q_{b},$$
where $\rho_{b}$, $C_{p,b}$, $k_{b,x}$, $k_{b,y}$, $k_{b,z}$, and $T_{b}$ represent the average density, specific heat capacity, effective thermal conductivity, and temperatures of the cell, respectively. t denotes time, while $Q_{b}$ is the rate at which the battery generates heat.

As the coolant flows through the LCP, it carries away the heat generated by the LBM. According to Newton’s cooling law, the convective heat transfer between the LCP and the fluid can be formulated as follows.
(5)$$q=hA_{1}(T_{w}-T_{l}),$$
where q is heat dissipated by convection, $T_{l}$ is the temperature of coolant, $T_{w}$ is the temperature of channel wall, h is convection heat transfer coefficient, and $A_{1}$ is convection heat transfer area.

On the basis of the heat generation model proposed by Bernardi [42], the heat (*Q*) generated by the battery is mainly the irreversible heat generated by the internal resistance of the battery and the reversible heat generated by the electrochemical reaction within the battery:(6)$$Q=I\left[\left(U_{OCV}-U\right)+T_{b}\frac{\partial U_{OCV}}{\partial T_{b}}\right],$$
where $I\left(U_{OCV}-U\right)$ is the Joule heat generated by the internal resistance of the battery. $IT_{b}\left(\partial U_{OCV}/\partial T_{b}\right)$ indicates the heat generated by an electrochemical reaction inside a battery. Therefore, the heating formula of the battery reads as follows:(7)$$Q=I^{2}R_{j}+IT_{b}\frac{\partial U_{OCV}}{\partial T_{b}}.$$

It is difficult to obtain accurate heat generation rates for individual cells in experiments. The following equation can be used to estimate the heat production of a single cell [43]:(8)$$Q=C_{p,b}m\Delta T,$$
(9)$$q=C_{p,b}m\frac{\partial T}{\partial t}.$$

The temperature rise data and the discharge time of the battery at different discharge rates are entered to calculate the heat generation of the battery at 1C, 2C, and 3C discharge rates, as shown in Table 3. 

In addition, the boundary conditions should be specified. All walls are adiabatic except for the upper surface. The thermal flux at the upper surface can be assumed as in Equation (9). Velocity inlet, a fixed temperature, and outflow at the outlet are assumed. No-slip conditions are enforced on all walls.

To evaluate the comprehensive cooling performance of LCP, theoretically, an index like performance evaluation criteria (*PEC*) should be defined.
(10)$$PEC=\frac{Nu}{f^{1/3}}.$$

However, Nusselts number under identical pump power in [18,19] is also usually employed. Popularly, the ratio of the power consumed to the heat taken away may be a more intuitive metric for cooling performance and is employed in the present work, where c and ρ are the specific heat and density of cooling fluid, respectively. ${\bar{T}}_{out}\;and\;T_{in}$ are average temperatures at the outlet and inlet, respectively. Δp is the pressure drop of LCP.
(11)$$\eta =\frac{c\left({\bar{T}}_{out}-T_{in}\right)}{\Delta p/\rho }.$$

### 3.2. Validity of the Numerical Model

In order to ensure the accuracy of simulation results, grid independence verification was carried out using the structure in Figure 3a. First, surface meshes of different cell sizes were generated, and then volume meshes were generated for numerical analysis under the same boundary conditions. The effect of the number of cell grids on the maximum temperature of the battery pack and the pressure drop of the liquid cooling system was tested. As can be seen in Figure 3, the maximum temperature decreased, and the relative error gradually decreased as the number of bulk grids increased. For the liquid-cooled system, the pressure drop tended to be constant as the number of meshes was encrypted. When the number of grids exceeded 6 million, the error in the maximum temperature and pressure drop of the LBM became small enough, so grid models with good convergence properties were selected for subsequent studies.

In order to validate the cell model, the simulation results were compared with the experimental data. Under 1C, 2C, and 3C discharge conditions, the heat powers of the battery were 3.24 W, 11.48 W, and 22.18 W, respectively. Discharge tests were conducted at 20 °C under different rates. The simulation tool STAR-CCM+ was used to obtain the changes in temperature, and the experimental temperature over time is depicted in Figure 4. It is clear that the simulation results closely match the experimentally measured temperatures, with a maximum error of approximately 2 °C. 

## 4. Results and Discussion

### 4.1. Cooling Effect of Different Channel Structures

SLCP and DLCP were arranged at the bottom of the LBM, and the battery pack was discharged at 1C under the environment of 20 °C. The initial temperature of coolant in the liquid cooling system was 20 °C, and the mass flow rate was 20 g/s. In Figure 5a,b, within 3600 s of discharge, the temperature of the battery pack increased with time and reached its highest value at the end of the discharge process. When the structure of the LCP was a serpentine channel structure, the maximum temperature of the battery pack was 24.05 °C; the maximum temperature of the double-layer channel structure was 23.85 °C. It was found that the DCLP slightly reduced the maximum temperature of the battery pack and improved the cooling performance of the liquid cooling system.

Figure 6 illustrates the maximum temperature and temperature difference for different cells in the LBM. The results demonstrate that the two different structures of the LCP can control the maximum temperature and temperature difference in the battery pack within a reasonable range. Additionally, the maximum temperature and the temperature difference between different cells in SLCP and DLCP exhibit a consistent trend. Specifically, the temperature was lower near the coolant inlet and gradually increased further away from the inlet. Furthermore, the maximum temperature of cells in DLCP was approximately 0.2 °C lower than that in SLCP, while the temperature difference between cells ranged from 0.1 °C to 0.2 °C. Therefore, the DLCP slightly reduced the temperature difference between the cells in LBM compared to SLCP.

The internal structure of the LCP has an impact on the flow of the coolant and subsequently affects the pressure drop in the liquid cooling system. Figure 7a illustrates the pressure distribution in two different flow channels. The pressure is highest at the channel entrance and lowest at the exit. It gradually decreases from the entrance to the exit, but the pressure drop is more evenly distributed in the DLCP.

Figure 7b illustrates the variation in pressure drop in the cooling system over time. The pressure drop curve exhibits an abrupt increase and decrease during the initial phase. This occurs when the LCP does not pass into the coolant, causing the pressure in each channel of the liquid cooling plate to be equal to the outlet pressure. As the coolant enters the initial phase of the cold plate, it does not immediately flow through all the channels, leading to a sharp rise in inlet pressure while the outlet pressure remains constant. Consequently, there is a rapid increase in pressure drops. As the coolant continues to flow in and the coolant channel extends, the contact area between the coolant and the channel increases, which leads to a decrease in flow resistance. Finally, the coolant flows through all channels, and the pressure drop of the cooling system is equalized. A remarkable observation is that the pressure distribution within each divergent channel of the DLCP structure appears to be uniform. In addition, the pressure drop in the two LCPs was measured to be 770 Pa and 519 Pa, which represents a reduction of 48% compared to the pressure drop in the SLCP structure. The main reason is that the inlet size for DLCP with one inlet is double that for SLCP with two inlets for obtaining the identical inlet velocity at a fixed total flow rate. A comparison of the comprehensive cooling performance defined by Equation (11) for DLCP and SLCP was conducted and indicates that cases with flowrates less than 20 g/s may have better cooling performance (Figure 8). Furthermore, all the findings suggest that DLCP exhibits improved cooling performance while consuming less power.

### 4.2. Cooling Effect at Different Discharge Rates

Figure 9 shows the temperature difference between the cells of the LBM and the temperature of the test site at different locations. During discharge at a rate of 1C, cell T8 is close to the coolant inlet and, therefore, has a slightly lower temperature, while cells T1 and T15 have slightly higher temperatures due to the distance between them. Throughout the LBM, the temperature difference between different cells at the same location during the discharge process is very small, indicating that the DLCP has good cooling performance and ensures uniform temperature between cells. In the vertical direction of the cell, the temperature is lowest for Tv1 and highest for Tv5. The temperature gradually increases from bottom to top, mainly because the closer the position is to the cold plate, the better the heat dissipation of the cell and the lower the temperature. It can be concluded that the temperature uniformity of the battery pack is better when the discharge rate is 1C, meeting the requirements of normal working conditions.

Under initial conditions, the battery pack was studied for 30 min to the end of discharge with a discharge rate of 2C. The simulation results are shown in Figure 10. In the case of a high discharge rate, the maximum temperature of the LBM rises to 31.61 °C, and the maximum temperature difference between cells is 6.41 °C. Although the temperature rises, the maximum temperature and average temperature between cells are 31.6 °C and 29.55 °C, respectively. The overall temperature uniformity of the LBM is good, and the temperature contours at the end of discharge are similar to that at 1C. The lowest temperature is slightly lower in the cells near the entrance. Due to the increase in the heat of the battery being large, the temperature difference in the vertical direction of the LBM gradually increases, and the heat is mainly concentrated in the upper part of the battery module, resulting in a large temperature difference between the upper and lower parts of the battery module.

### 4.3. Effect of Coolant Flow Rate and Temperature

In the previous section, the initial condition of the coolant was constant. In order to explore the influence of the coolant on the cooling performance, the flow rate and temperature should be changed to explore the cooling effect.

#### 4.3.1. Effect of Coolant Flow Rate

Under the discharge rate of 2C, the temperature of the coolant was maintained at 20 °C, and the cooling effect was studied when the mass flow rate was 10 g/s, 20 g/s, 30 g/s, 40 g/s, and 50 g/s. As shown in Figure 11a, with the increase in mass flow rate, the cooling performance was improved to some extent. The maximum temperature and minimum temperature of the LBM decreased gradually with the increase in the mass flow rate. When the mass flow rate reached 30 g/s, the maximum temperature and minimum temperature of the LBM changed gently with the increase in the mass flow rate, and the maximum temperature was effectively controlled within 32 °C. However, the temperature difference in a battery cell gradually increased with the increase in mass flow rate. This is mainly due to the limited thermal conductivity of the cell, which prevents the heat at the top of the cell from being transferred to the bottom in time, resulting in a lower temperature at the bottom and a higher temperature at the top.

The energy consumption of the liquid cooling system increased as the mass flow rate increased. Figure 11b shows the pressure drop of the liquid cooling system. Therefore, simply increasing the flow rate does not improve the cooling performance. The liquid cooling system exhibited better cooling performance when the mass flow rate was 20 g/s. At this flow rate, the maximum temperature of the LBM was 31.81 °C, the temperature difference in the battery module was 6.35 °C, and the average temperature of the monomer was approximately 29.5 °C. Additionally, the energy consumed was relatively low.

#### 4.3.2. Effect of Coolant Temperature

The initial temperature of the coolant, battery, and ambient temperature was to 20 °C. When the LBM was discharged at the rate of 2C, the inlet temperature of the coolant was set as 15 °C, 20 °C, and 25 °C, respectively, and the mass flow rate was 20 g/s.

As shown in Figure 12, it can be found that the maximum temperature of the cell is relatively uniform for the three different coolant temperatures, but the temperature difference between cells near the coolant inlet is large. When the inlet temperature of the coolant was 15 °C, after 30 min of discharge, the maximum temperature of the battery module was 27.96 °C. The maximum temperature difference between cells was 7.06 °C. When the initial coolant was 20 °C, the maximum temperature of the battery module increased by 3.65 °C. The maximum temperature difference between monomers was 6.35 °C. When the coolant was 25 °C, the maximum temperature of the battery module was 35.07 °C, and the temperature difference between cells was reduced by 0.69 °C. Obviously, the lower the inlet temperature of the coolant, the better the cooling performance of the liquid cooling system and the lower the maximum temperature control. However, the lower the temperature at the inlet, the lower the temperature at the bottom of the LBM for a longer period of time, and the high temperature at the top cannot be transferred to the bottom in a timely manner, resulting in a larger temperature difference between cells. When the inlet temperature of the coolant was 20 °C, which is close to the initial temperature of the battery module, the maximum temperature of the battery pack and the temperature difference between cells were well controlled.

Finally, a comparison of the cooling efficiency for different discharge rates and fluid temperatures at the inlet was performed. The results in Figure 13 indicate that the cooling efficiency at 2C and 15° was the highest. Since less heat can be emitted at a lower discharge rate, a larger flow rate or a lower fluid temperature is not required for cooling requirements.

### 4.4. Effect of Adding Auxiliary Cold Plate

To address the large temperature difference in the LBM during high-rate discharges, we arranged the auxiliary LCPs on the side of the module to form a double-layer minichannel at the bottom and an auxiliary flow channel on the side. According to Figure 2, the addition of auxiliary LCPs to the liquid cooling system resulted in a mass flow rate of 10 g/s for the bottom channel and 5 g/s for the side channel, while the mass flow path remained at 20 g/s. Figure 14 shows that at a discharge rate of 2C, the maximum temperature of LBM in the DLCP with auxiliary LCPs was 27.86 °C after 30 min of discharge, which was 3.75 °C lower than DLCP. Additionally, the maximum temperature difference in monomer in the module was 4.09 °C, which is 2.26 °C lower than that of DLCP. These results indicate that the addition of auxiliary LCPs improves the cooling performance of the liquid cooling system and reduces the maximum temperature of LBM. It also ensures the uniformity of battery temperature under high-rate discharge conditions.

### 4.5. Effect of Non-Constant Discharge

During the actual application process, the discharge rate of the battery dynamically changed with different power consumption of the device. As shown in Figure 15, three dynamical simulation tests with different discharge rates were designed to study the cooling effect of the liquid cooling system. In test 1, the battery discharge rate first increased and then decreased. In test 2, the battery discharge rate was set alternately from small to large. In test 3, the battery discharge rate varied from small to large, and the rest time was set between the different rates.

Figure 16a shows that the maximum temperature of the LBM changes with time when no coolant works. The rate of temperature rise varied between the three experiments. In Figure 16b, test 1 shows that the temperature rise rate of the battery gradually increased with the increase in the discharge rate, and the temperature difference between the cells increased with the increase in the maximum temperature. When the maximum temperature reached a certain value, the temperature rise rate decreased with decreasing discharge rate. Due to the temperature gradient inside the cell, the temperature of the battery string started to decrease when the heat dissipation capacity of the liquid cooling system was higher than the heat generation power of the battery. In Figure 16c, test 2 shows the discharge rate alternated between 3C and 1C; it can be found that the temperature of the LBM decreased when the discharge was at 1C. However, at high temperatures, the rate of temperature decrease was significantly higher than at low temperatures due to the large temperature difference. In Figure 16d, after discharging 150 s at 1C, the battery stagnated for 100 s. Due to the low-temperature rise of the cell, the temperature difference between cells was small, and the temperature decreased slowly. The cooling rate of the cell temperature increased significantly in the standing state, indicating that the larger temperature difference in the cell led to a rapid drop in the cell temperature when the cell was not in operation.

Compared with the constant discharge experiment, the non-constant discharge experiment better reflects the dynamic changes in battery temperature with different discharge powers. The data analysis from the monitoring points shows that the maximum temperature of the cell was close to the value of Tv5, and the average temperature of the cell was not much different from the value of Tv3. Setting the monitoring points at these two locations accurately reflects the dynamic changes in the battery temperature.

## 5. Conclusions

In this paper, a new SLCP and DLCP were designed, and two LCP layout modes were proposed in light of the requirements raised by a factory. The cooling performance of SLCP and DLCP under the same initial conditions was investigated. The cooling effect of the DLCP was investigated for different discharge rates, the influence of coolant flow and temperature on the cooling effect was controlled, and the dynamic change in temperature at different monitoring points during non-constant current discharge was explored. The conclusions are as follows:(1)When the discharge rate was 1C, the maximum temperature and temperature difference in the DLCP were 0.2 °C and 0.1 °C lower than those of the SLCP, respectively. The pressure drop of the DLCP liquid cooling system was 26% lower than that of the SLCP system under the same initial conditions. The comparison indicates that DLCP has better comprehensive cooling performance compared to SLCP.(2)Increasing the coolant flow rate improved the cooling performance. When the mass flow rate reached 30 g/s, the maximum temperature of the cell and the temperature difference tended to vary gently. Considering the energy consumption, a mass outflow rate of 20 g/s is more appropriate. Reducing the temperature of the coolant effectively reduced the maximum temperature of the LBM but caused a large temperature difference between cells. Therefore, the temperature of the coolant should be controlled to be close to the initial temperature of the battery, which is 20 °C.(3)When the battery discharge rate was 2C, the liquid cooling system with an auxiliary cold plate effectively controlled the maximum high temperature of the LBM within 28 °C, and the temperature difference between the cells was maintained at approximately 4 °C. In the case of high-rate discharges, the cooling performance of the liquid cooling system can be effectively improved by increasing the contact area between the LCP and the LBM.(4)When the current was not constant, the temperature difference in the cell increased with the increase in temperature. When the heating power of the cell was reduced, the larger internal temperature difference in the cell at higher temperatures favored a rapid decrease in cell temperature.

## Figures and Tables

**Figure 1 micromachines-14-02128-f001:**
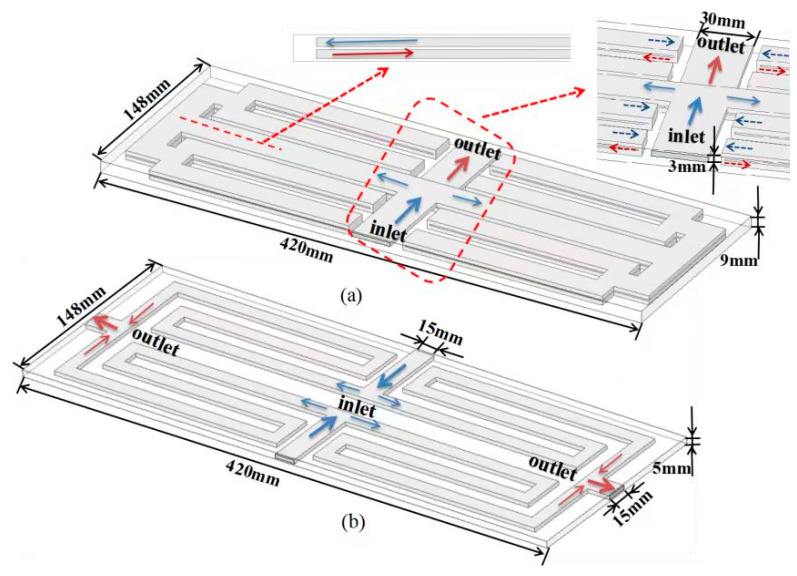
The layout of the liquid cooling plate: (**a**) bottom liquid cooling plate and (**b**) bottom and side auxiliary liquid cooling plate.

**Figure 2 micromachines-14-02128-f002:**
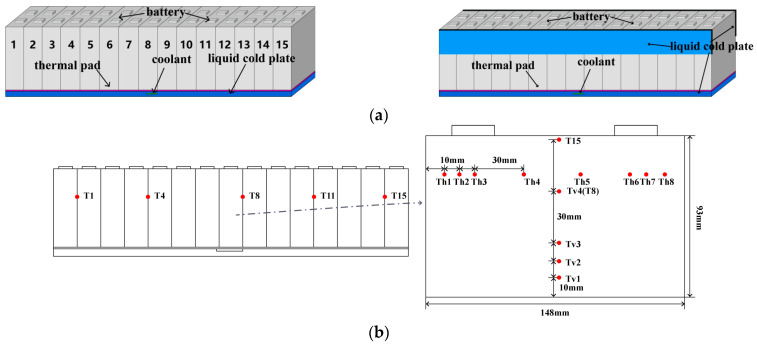
(**a**) The LBM with bottom and side auxiliary LCPs and (**b**) the temperature monitoring points.

**Figure 3 micromachines-14-02128-f003:**
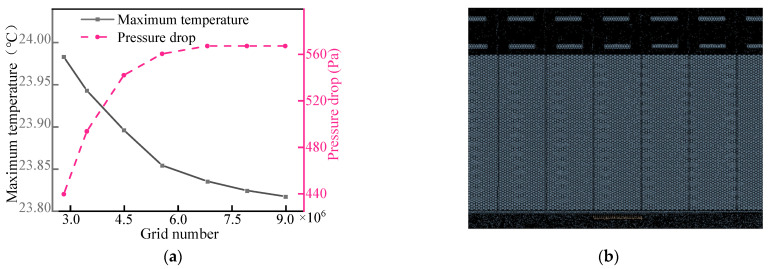
(**a**) Validity of grid independence and (**b**) the grids of the simulation domain.

**Figure 4 micromachines-14-02128-f004:**
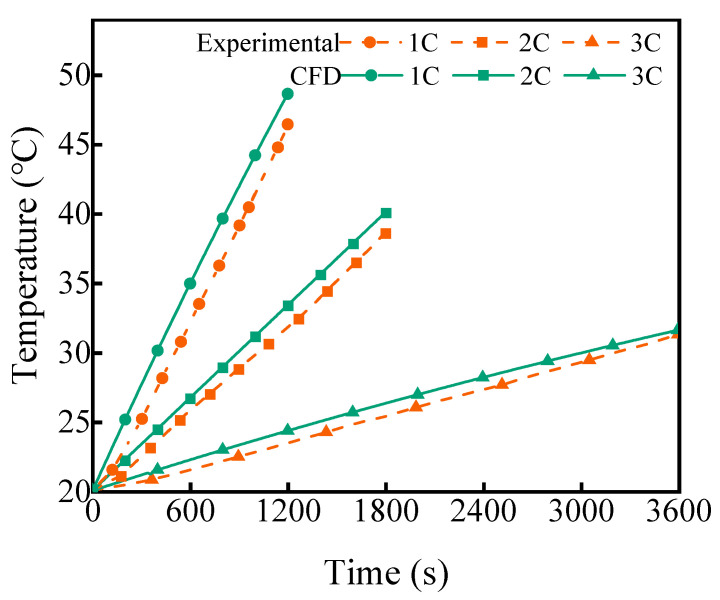
CFD and experimental comparison of temperature rise of battery cell with different discharge rates Ref. [39].

**Figure 5 micromachines-14-02128-f005:**
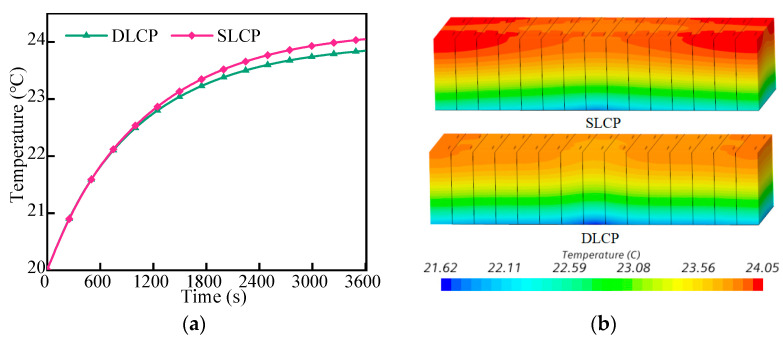
Maximum temperature change of 1C discharge rate (**a**) and the temperature contours of the battery module (**b**).

**Figure 6 micromachines-14-02128-f006:**
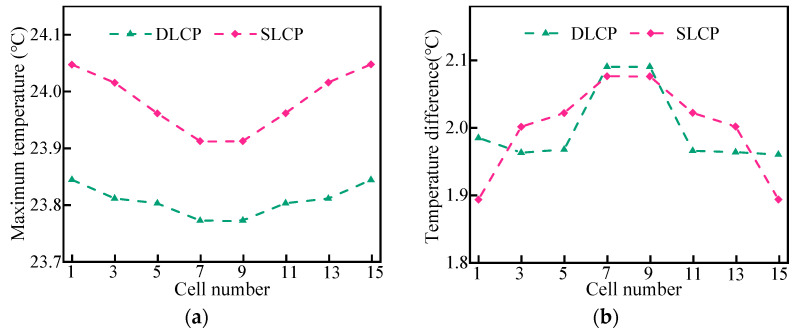
The temperature of cells with different channel structures: (**a**) maximum temperature and (**b**) temperature difference.

**Figure 7 micromachines-14-02128-f007:**
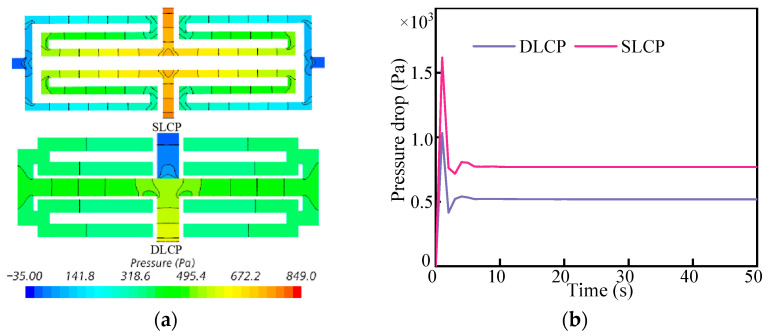
The pressure contours with different channel structures (**a**) and evolution of pressure drop with channel structure (**b**).

**Figure 8 micromachines-14-02128-f008:**
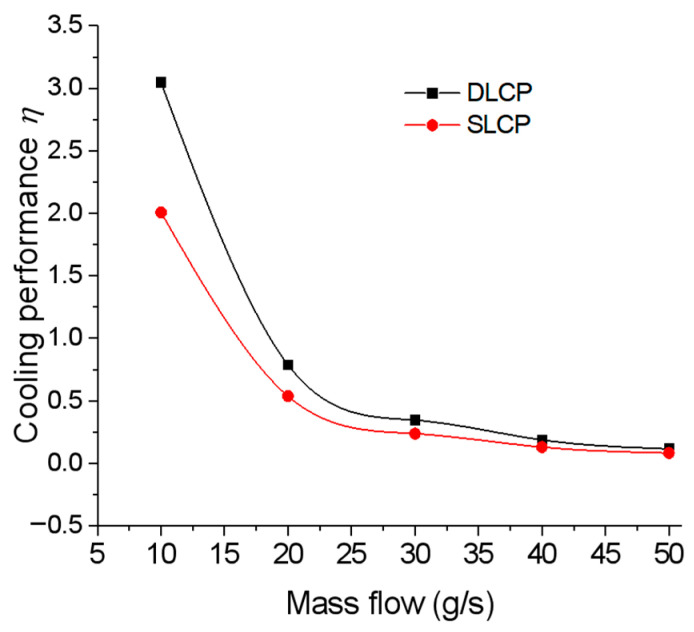
Comparison of the thermal performance of DLCP and SLCP for 2C and 20 g/s.

**Figure 9 micromachines-14-02128-f009:**
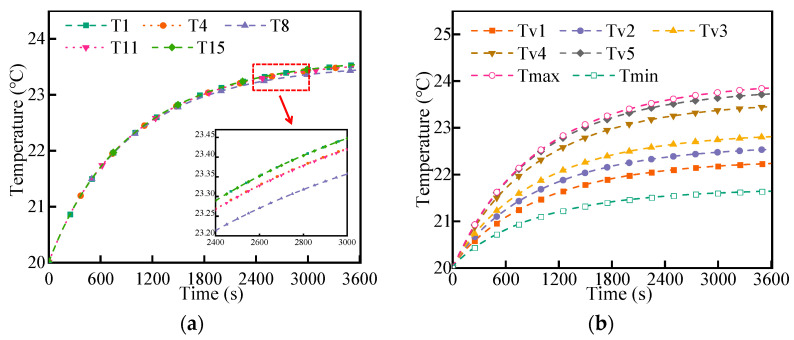
C Discharge temperature: (**a**) temperature between different cells and (**b**) temperature on the surface of the cell.

**Figure 10 micromachines-14-02128-f010:**
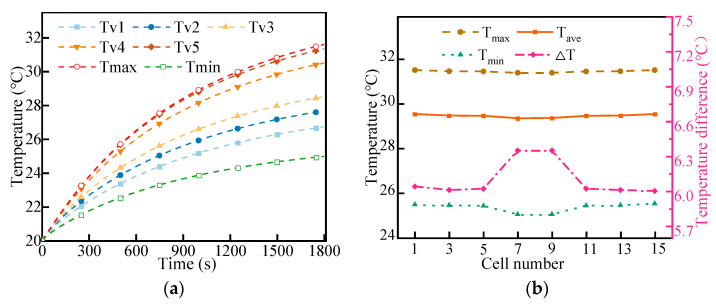
2C Discharge temperature: (**a**) temperature rise curves of the monitoring points and (**b**) temperature between the different cells.

**Figure 11 micromachines-14-02128-f011:**
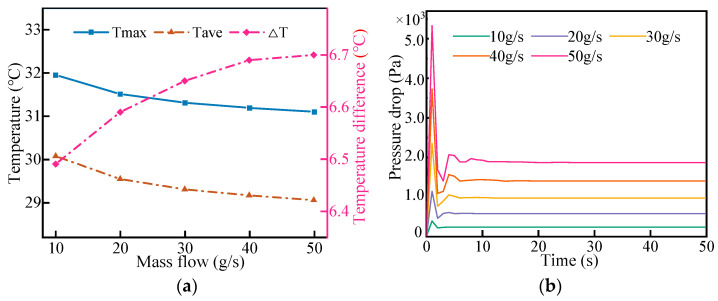
Effects of different flow rates on thermal management performance (**a**) and evolution of pressure drop with mass flow rate (**b**).

**Figure 12 micromachines-14-02128-f012:**
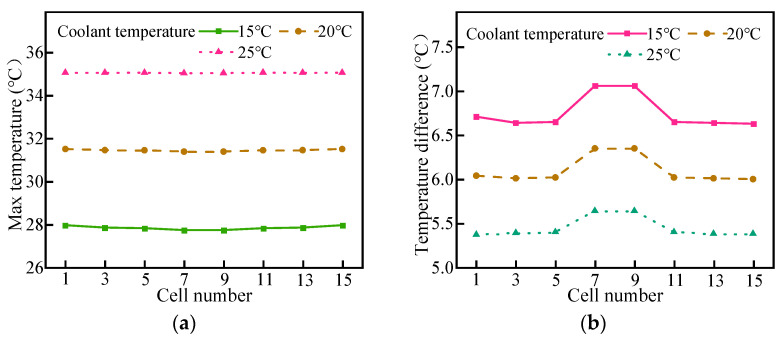
Effects of inlet coolant temperature: (**a**) maximum temperature and (**b**) temperature difference.

**Figure 13 micromachines-14-02128-f013:**
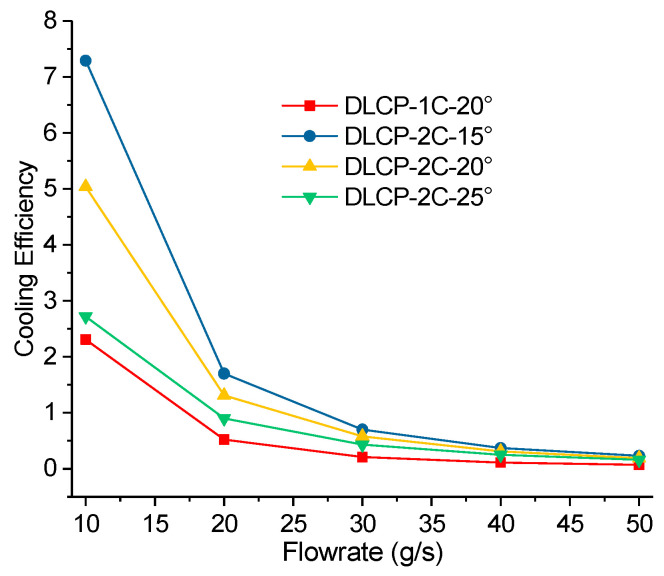
Comparison of the thermal performance of DLCP at different fluid temperatures and discharge rates.

**Figure 14 micromachines-14-02128-f014:**
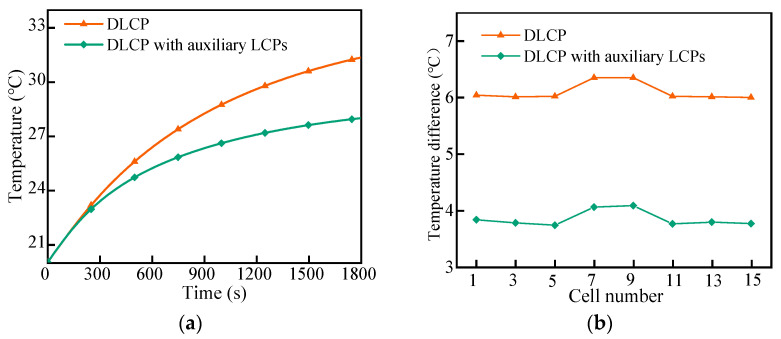
Effect of adding auxiliary cold plate: (**a**) vertical temperature and (**b**) horizontal temperature.

**Figure 15 micromachines-14-02128-f015:**
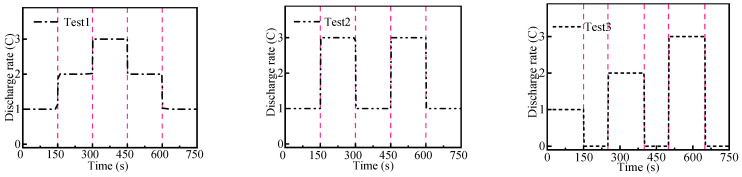
Three experiments with non-constant currents.

**Figure 16 micromachines-14-02128-f016:**
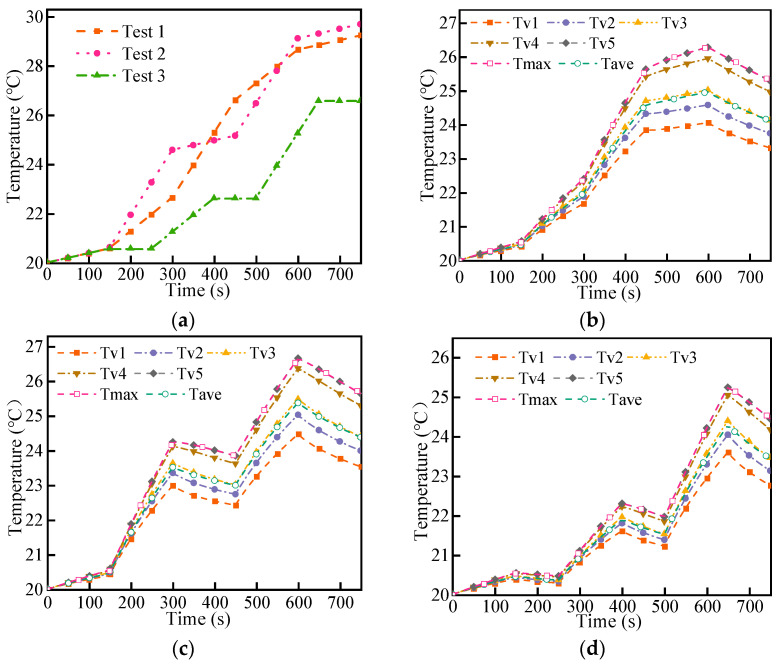
Temperature of battery versus time: (**a**) temperature without coolant, (**b**) temperature of test 1, (**c**) temperature of test 2, and (**d**) temperature of test 3.

**Table 1 micromachines-14-02128-t001:** Parameters of lithium-ion battery Ref. [39].

Specification	Parameters
Nominal capacity (Ah)	40
Nominal voltage (V)	3.6
Mass (g)	825 ± 10
Dimensions (mm)	148 × 93 × 28 (length × width × thickness)
Density (kg/m^3^)	2100
Specific heat (J/kg·K)	1030
Thermal conductivity (W/(m·K))	18.7/18.7/1.8
Preferences for inlet and outlet	One in and one out

**Table 2 micromachines-14-02128-t002:** Material properties in the simulations Ref. [39].

	Aluminum	50/50 Glycol Water	Thermal Pad
Density (kg/m^3^)	2719	1065	1200
Specific heat (J/kg·K)	871	3494	800
Thermal conductivity (W/(m·K))	202	0.419	2
Dynamic viscosity (Pa.s)	—	0.0035	

**Table 3 micromachines-14-02128-t003:** Heat generation of battery under different discharge rates.

Discharge rate	1C	2C	3C
Heat generation	3.24 w	11.48 w	22.18 w

Note that the mass flow is set to be 20 g/s, and the initial temperature of the coolant and the battery are 20 °C at inlet. A pressure outlet was set at the outlet.

## Data Availability

Data are contained within the article.

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
