# Peer review of "Numerical Study on a Liquid Cooling Plate with a Double-Layer Minichannel for a Lithium Battery Module"

_micromachines, 2023, doi:10.3390/mi14112128_

Round 1
Reviewer 1 Report
Comments and Suggestions for Authors
This paper discusses what might be a novel cooling approach for batteries using a so-called double-layered-structured cooling plate instead of a single-layered structure. This appears to offer some reduction in pressure drop and an unclear or marginal improvement of thermal performance. The paper has major problems that need significant revision. Unless these problems are addressed, this paper should be rejected.
*Major concerns*
DLCP offers a <1% enhancement of maximum temperature reduction in comparison with SLCP (Figs. 5, 6a). So you cannot conclude that "the LCP with double-layer minichannel structure has better cooling performance". You would have to say something like "the LCP with double-layer minichannel structure has *marginally* better cooling performance". Mention this in both the conclusion and abstract. Please do not misguide your reader.
Stress in the following that the difference of maximum temperature in the two designs is minor. "When the structure of the LCP is serpentine channel structure, the maximum temperature of the battery pack is 24.05℃; the maximum temperature of the double-layer channel structure is 23.85℃. It can be found that the DCLP effectively reduces the maximum temperature of the battery pack and improves the cooling performance of the liquid cooling system."
DLCP offers about 50% enhancement in pressure drop in comparison with SLCP (Fig 7). Explain why. Is it because the channel cross-sectional area is smaller in one of the two designs?
This pressure-drop performance, unlike the above-mentioned thermal performance, looks like a significant improvement. Is this the main contribution of your work? If yes, why didn't you mention this in your conclusions and abstract? Please clarify this issue.
The paper says this regarding Fig 12: "By analyzing the data from the vertical and horizontal monitoring points, it was found that the addition of the side-assisted cold plate improves the cooling performance of the liquid cooling system and reduces the maximum temperature of the LBM." This reviewer cannot see this from your Fig 12. So please redo this figure, or re-analyze your results.
This paper shows no validation with experiments. This is OK since problems such as the one considered in this paper are known to be well captured by CFD. But you must show this to the reader with validation for similar heat transfer problems to the one in this paper. You can do this validation yourself. Or you could show the work done by others. In the latter case, you will need to redo figures from other authors yourself and adapt them to your work, or use figures from past papers and ask for permission to corresponding journals.
*Minor concerns*
The last paragraph of the introduction states your goals. Please clarify how these goals are different than those of previous papers. In other words, explain what makes this paper novel.
You seem to be conducting a conjugate heat transfer calculation. If yes, please say so, and mention which CFD code you used.
Please put this document through a good grammar checker. Spelling, grammar, and punctuation all need to be checked for.
Correct "E et al.[28]".
Figures 1 and 2 are the same. Fix this.
Comments on the Quality of English LanguageSee my previous comments.
Author Response
See in attachment.

Reviewer 2 Report
Comments and Suggestions for Authors
In this work, the authors numerically studied the thermal performances of a liquid cooling plate with double-layer minichannel for the cooling of Lithium Battery Module. The present work is important to reduce the work temperature and ensure the performance of lithium battery module. However, the results presented in the manuscript need significant improvement by addressing the following comments.
(1) For the current numerical study, although the authors conducted the grid independence test to ensure the accuracy of the numerical results, there is a lack of essential data validation with the literatures, especially the data comparison with experimental literatures.
(2) The present work mainly focused on thermal performances of the liquid cooling plate. We all know that the improvement of the thermal performances consumes more energy, which means increasing pressure loss. Thus, the authors need discuss the pressure loss of their designs. At the same time, the thermal performance factor denoted as η, which is defined as follows must be considered to analyze the comprehensive hydraulic and thermal performance of the liquid cooling plate.
η=(Nu/Nuref)/(Δp/Δpref)1/3
(3) The latest literatures related to the current topic should be cited.
Comments on the Quality of English Language
Minor editing of English language required
Author Response
See in attachment.

Reviewer 3 Report
Comments and Suggestions for Authors
1. For the model validation, the simulation results by the present model should be compared with the experimental data. However, the author only presents the grid independence test, which make the results not solid.
2. For the simulation, which software was used? It should be declared.
3. The boundary conditions should be given for the simulation cases.
Comments on the Quality of English LanguageMinor editing of English language required
Author Response
See in attachment.

Round 2
Reviewer 1 Report
Comments and Suggestions for Authors
The paper now looks much better. But you must make one minor change. Please mention *in the paper* that you conducted conjugate heat transfer simulations with Siemens StarCCM+, as you indicated in your response. This will strengthen your paper since StarCCM+ is a very good code.
Comments on the Quality of English LanguageJust put the paper through a spell and grammar checker one last time.
Author Response
Thanks for your positive comments.
We have specified the CFD software, STAR-CCM+, in revised manuscript.
Reviewer 2 Report
Comments and Suggestions for Authors
Although the authors address some of my comments, the manuscript still need further improvement.
(1) Although the authors added some discussion on the cooling performance, it is still not sufficient. The cooling performance should be discussed for each influencing factor they studied.
Comments on the Quality of English LanguageModerate editing of English language required
Author Response
Although the authore added some discussion on the cooling performance, it is still not sufficient. The cooling performance should be discussed for each influencing factor they studied.
Response: Thanks for your suggestion. We added a comparison of colling efficiency for various flowrates, various temperatures at inlet and various discharge rates in Figure 13.
Reviewer 3 Report
Comments and Suggestions for Authors
The suggestions mentioned previously are addressed, it could be accepted.
Author Response
Thanks for your positive comment.
Round 3
Reviewer 2 Report
Comments and Suggestions for Authors
Accept
Comments on the Quality of English LanguageNo comments